# Essential Oil Composition Analysis of *Cymbopogon* Species from Eastern Nepal by GC-MS and Chiral GC-MS, and Antimicrobial Activity of Some Major Compounds

**DOI:** 10.3390/molecules28020543

**Published:** 2023-01-05

**Authors:** Sabita Dangol, Darbin Kumar Poudel, Pawan Kumar Ojha, Salina Maharjan, Ambika Poudel, Rakesh Satyal, Anil Rokaya, Sujan Timsina, Noura S. Dosoky, Prabodh Satyal, William N. Setzer

**Affiliations:** 1Analytica Research Center, Kirtipur, Kathmandu 446088, Nepal; 2Aromatic Plant Research Center, 230 N 1200 E, Suite 102, Lehi, UT 84043, USA; 3Department of Chemistry, University of Alabama in Huntsville, Huntsville, AL 35899, USA

**Keywords:** citronella, palmarosa, lemongrass, citral, enantiomeric distribution, antibacterial, antifungal

## Abstract

*Cymbopogon* species essential oil (EO) carries significant importance in pharmaceuticals, aromatherapy, food, etc. The chemical compositions of *Cymbopogon* spp. *Viz. Cymbopogon winterianus* (citronella) *Cymbopogon citratus* (lemongrass), and *Cymbopogon martini* (palmarosa) were analyzed by gas chromatography–mass spectrometry (GC-MS), enantiomeric distribution by chiral GC-MS, and antimicrobial activities of some selected pure major compound and root and leaves EOs of citronella. The EO of leaves of *Cymbopogon* spp. showed comparatively higher yield than roots or other parts. Contrary to citral (neral and geranial) being a predominant compound of *Cymbopogon* spp., α-elemol (53.1%), α-elemol (29.5%), geraniol (37.1%), and citral (90.4%) were detected as major compounds of the root, root hair with stalk, leaf, and root stalk with shoot of citronella EO, respectively. Palmarosa leaves’ EO contains neral (36.1%) and geranial (53.1) as the major compounds. In the roots of palmarosa EO, the prime components were α-elemol (31.5%), geranial (25.0%), and neral (16.6%). Similarly, lemongrass leaves’ EO contains geraniol (76.6%) and geranyl acetate (15.2%) as major compounds, while the root EO contains a higher amount of geraniol (87.9%) and lower amount of geranyl acetate (4.4%). This study reports for the first time chiral terpenoids from *Cymbopogon* spp. EOs. Chiral GC-MS gave specific enantiomeric distributions of nine, six, and five chiral terpenoids in the root, root stalk with a shoot, and leaves of citronella EOs, respectively. Likewise, four and three chiral terpenoids in the root and leaves of lemongrass oil followed by two chiral terpenoids in the leaves and root of palmarosa EOs each. Additionally, the root and leaves’ EOs of citronella exhibit noticeable activity on bacteria such as *Staphylococcus aureus*, *Staphylococcus epidermidis*, and *Streptococcus pyogenes* and fungus such as *Candida albicans*, *Microsporum canis*, and *Trichophyton mentagrophytes*. So, geranial-, neral-, geraniol-, and citronellal-rich EOs can be used as an alternative antimicrobial agent.

## 1. Introduction

The genus *Cymbopogon* (Poaceae) comprises up to 144 species distributed in Asia, America, and Africa [1,2], among which *Cymbopogon winterianus* Jowitt ex Bor, *Cymbopogon citratus* Stapf, and *Cymbopogon martini* (Roxb.) Will. Watson are the economically important species of the genus [3]. The species *C. winterianus* is popularly known as citronella grass, citronella, or Java citronella [4]. Citronella is one of the industrially important essential oils (EO), which is produced by steam or hydro-distillation of the whole plant [5] and has a characteristic lemon odor [6]. The world consumption of citronella EO has amounted to several thousand tons annually due to being the most important source of geraniol and citronellal [7]. It is also known for its natural insect-repellent property and is of great interest to the pharmaceutical and fragrance industry [8]. It has various therapeutic uses as an analgesic, anticonvulsant, anxiolytic, etc., and is a favorable agent for antifungal, antibacterial, antiparasitic, herbicidal [9], and nematicidal activities. In traditional practice, citronella EO has been used as an antipyretic, aromatic tea, vermifuge, diuretic, and in treating mental illness [10,11,12,13,14].

*Cymbopogon citratus*, also known as lemongrass [15], is widely cultivated worldwide due to its aromatic and medicinal properties [16]. It is rich in minerals, vitamins, and macronutrients (carbohydrates, protein, and small amounts of fat). Its leaves are also a good source of various bioactive compounds, including alkaloids, terpenoids, flavonoids, phenols, saponins, and tannins [17], which has led to numerous herbal therapies currently widely used in medicine [18]. Lemongrass has been traditionally used as a remedy for a variety of health conditions. Recent scientific studies have provided evidence supporting its antiviral, antimicrobial [19], antioxidant [20], antifungal [21], anticancer [22], sedative, and anti-inflammatory properties in several disease models [23,24]. It is used as a food flavoring and as a perfume and contains 1 to 2% EO on a dry weight basis [25]. The quality of lemongrass is generally determined by the amount of citral content, which is composed of isomers (neral and geranial), giving it the characteristic odor [26].

*Cymbopogon martini* (palmarosa) is an important, multiharvest, perennial, ad aromatic grass native to India. India is the principal producer and exporter of palmarosa EO [27,28,29]. EO of palmarosa is generally distilled (more frequently by steam or hydro-distillation) from the leaves, stems, flowers, bark or roots, or other elements of the plant [30]. The EO of palmarosa mainly consists of monoterpenoid components [31] such as citral, citronellol, and geranyl acetate [32], which is why these oils are mostly used as a culinary flavoring in Asia [33,34,35] and have demonstrated remarkable bioactivities, including herbicidal activity [36]. Moreover, the most important use of palmarosa is in the treatment of skin infections such as acne and also to stimulate cell regeneration while regulating the production of sebum [37].

EOs are expensive because of their low oil yields. In order to increase profit margins, EOs are being adulterated in different degrees with vegetable oils, synthetic compounds, or isolated natural compounds from less valued EOs [38]. So, there is a dire need to understand the composition of *Cymbopogon winterianus*, *Cymbopogon citratus*, and *Cymbopogon martini*. Thus, in the present study, we isolated the EOs from different tissues of these three *Cymbopogon* species and studied their composition using GC-MS and Chiral GC-MS, which can be used as a chemical fingerprint for the authentication of these *Cymbopogon* species. In addition to these, we studied the antibacterial and antifungal activity of some pure major components (geraniol, (±) citronellol, citral, and (−) citronellal) and the root and leaves of *Cymbopogon winterianus* Eos.

## 2. Results and Discussion

### 2.1. Essential Oil Yields

The EO yields of three *Cymbopogon* species, namely *C. winterianus*, *C. citratus*, and *C. martini* are presented in Table 1. The hydro-distillation of leaves of all three *Cymbopogon* species yielded 1.3–2.2% essential oil, which is remarkably high compared with other parts of the plant. The previously reported hydro-distillation yield of *C. winterianus* was 1.50 ± 0.15% (*w/w*) and with better extraction yield in comparison with steam distillation [39]. Similarly, the hydro-distillation extraction yield of *C. citratus* was 0.98%, which was quite lower as compared with microwave-assisted hydro-distillation. Our extraction yield was closely in agreement with previously reported microwave-assisted hydro-distillation yields [40]. In the case of *C. martini* leaves, the yield of extraction was (1.4756 *w/w* %), which is in close agreement with our hydro-distillation yield [41].

### 2.2. Chemical Composition of Cymbopogon Species

The major compounds present in the different tissues of *cymbopogon* species are presented in Figure 1. Several studies have been reported on the EO composition of *C. winterianus*, which reveals high variability in its chemical fingerprint. So, to reveal the mystery behind this variability for the first time, we studied the composition of EOs extracted from different parts of *C. winterianus*. The results of GC-MS analysis of *C. winterianus* EOs from different parts are presented in Appendix A, and Table 2 shows only the major selected constituents.

In the root oil of *C. winterianus*, sesquiterpenoids were predominant, representing 84.5% of the total oil in which monoterpenoids were 9.8% and sesquiterpenes 4.7%. The result shows that the roots were rich in α-elemol (53.1%), α-eudesmol (18.9%), γ-eudesmol (7.5%), whereas the root hair and stalk oils are dominated by more than half by the monoterpenoids (52.1%), sesquiterpenoids (41.8%), and sesquiterpenes (3.1%) of the total oil. The principal compound of root hair and stalk was α-elemol (29.5%), geranial (10.7%), citronellal (8.6%), geraniol (8.6%), citronellol (7.5%), neral (6.5%), and geranyl acetate (5.3%). The composition of root and root hair and stalk oils are almost similar, presenting α-elemol as the major compound. The percentage of some compounds such as citronellol, citronellal, neral, geraniol, geranial, and geranyl acetate are higher in root hair and stalk EO, while, other compounds such as α-elemol, α-Eudesmol, and γ-Eudesmol are higher in the root oil of *C. winterianus*. 

The leaves and root stalk and shoot oils largely contained monoterpenoids as their major components. The leaves’ EO contains (90.4%) monoterpenoids, (3.9%) sesquiterpenes, and (2.6%) sesquiterpenoids. The major compounds of leaves were geraniol (37.1%), followed by citronellal (13.7%), geranyl acetate (11.3%), geranial (11.0%), neral (8.3%), and citronellol (5.1%); this was in a close agreement with studies conducted by Rodrigues et al. [4], except the presence of α-elemol in high amounts, which might be due to the addition of roots in their studies during EO extraction. In another study, it is unclear whether the part used in their extracted EO is either the leaves, stems or both of *C. winterianus*; anyways, the principle components are citronellal, geraniol, and citronellol [44]. Similarly, contrasting the type of *C. winterianus* leaves EO composition and major component was geraniol (18.88%), citronellal (16.95%), elemenol (14.08%), and citronellol (12.57%), and we confirm that was not pure single-tissues EO, and there might be addition of other tissues [45]. Meanwhile, 94.2% monoterpenoids, 1.8% sesquiterpenoids, and 0.5% sesquiterpenes were found in root stalk and shoot EO, where citral was the predominant constituent, consisting of geranial (59.0%) and neral (31.5%).

The results of GC-MS analysis of *Cymbopogon citratus* EOs extracted from roots and leaves are tabulated in Appendix A, and Table 3 shows only the selected major constituents.

*C. citratus* leaves’ EO contains neral (36.1%) and geranial (53.1%) monoterpene aldehydes as the major compounds, whereas (*E*)-β-caryophyllene (1.0%) and caryophyllene oxide (1.3%) contain γ-cadinene (0.81%) in lower concentrations, which is similar to the study previously conducted by Boukhatem et al. [18], except for the presence of β-myrcene in a significant amount. *C. citratus* leaves’ EO from Vietnam also showed a comparable content of citral (neral and geranial), and the content of β-myrcene can be used to distinguish the origin of the EO [46]. In the roots of *C. citratus* EO, a sesquiterpene alcohol and monoterpene aldehydes were predominant compounds. The prime components were α-elemol (31.5%), geranial (25.0%), neral (16.6%), α-eudesmol (11.3%), and γ-eudesmol (5.2%); *trans*-β-elemene (1.4%) was present in a lesser quantity. Thus, compounds such as α-elemol, α-eudesmol, and γ-eudesmol can be used to identify the source of *C. citratus* EO. 

The results of GC-MS analysis of *Cymbopogon martini* EOs extracted from leaves and roots are presented in Appendix A, and Table 4 shows only the selected constituents. The *C. martini* leaves’ EO contains geraniol (76.6%), and geranyl acetate (15.2%) as major compounds, which is comparable to the previous study conducted by Jnanesha et al. [27]. Geraniol content in the *C. martini* leaves’ EO up to 80 days was noticeably increased, whereas geranyl acetate decreased significantly at that time and positively correlated [3]. Interestingly, one of the *C. martini* cultivars from India showed limonene as a major compound [47], while, the *C. martini* root EO contains a higher amount of geraniol (87.9%) compared with the leaves’ oil, which ultimately leads to a decrease in the content of geranyl acetate (4.4%). The underlying reasons for the differences in EOs composition could be attributed to genotype, edaphic variables, geographical location, pedo-climatic conditions, harvest time, extraction procedure, maturity of plant, different part of plant material, and analytical procedures.

### 2.3. Enantiomeric Distributions Analysis of Cymbopogon Species Essential Oils

The enantiomeric distributions of chiral terpenoids present in *Cymbopogon* species EOs are presented in Table 5. In previous studies, the enantiomeric distributions of chiral terpenoids have been successfully used for species identification and adulteration detection of different EOs [38,48]. To the best of our knowledge, this is the first report on enantiomeric distributions of chiral terpenoids from *Cymbopogon* species *Viz C. winterianus*, *C. citratus*, and *C. martini*. There were, altogether, nine chiral terpenoids detected in various parts of citronella Eos, among which linalool, terpinen-4-ol, bornyl acetate, borneol, α-terpineol, and (*E*)-β-caryophyllene were levorotatory. However, citronellal and citronellol were detected as dextrorotatory compounds. On the contrary, germacrene D was levorotatory in the root and dextrorotatory in leaves, as well as root stalk and shoots.

There were, altogether, five chiral terpenoids detected in lemongrass root and leaves EOs. Among these, linalool and (*E*)-β-caryophyllene were levorotatory and detected in both root and leaf oil. However, citronellol and citronellal were dextrorotatory and detected only in root EO. Germacrene D, on the other hand, was dextrorotatory but detected only in the lemongrass leaves’ oil. Lastly, only two chiral terpenoids were detected in palmarosa root and leaves’ EOs, namely linalool and (*E*)-β-caryophyllene; linalool was dextrorotatory and (*E*)-β-caryophyllene was 100% levorotatory.

Thus, the enantiomeric distributions of chiral terpenoids present in *Cymbopogon* species *Viz C. winterianus*, *C. citratus*, and *C. martini* will be helpful to establish the chemical fingerprint of these species and also in the adulteration detection of EOs of these species.

### 2.4. Antimicrobial Activity of Cymbopogon Winterianus Essential Oil and Some Major Compounds

The MIC values of *Cymbopogon winterianus* EOs and some pure compounds such as geraniol, (±) citronellol, citral, and (−) citronellal, and that of the positive control gentamicin against a panel of bacterial and fungal strains, were determined through a two-fold broth microdilution method. This study showed that the assayed root and leaves of *C. winterianus* EOs have variable microbial inhibitory activities, as presented in Table 6. Plants having secondary metabolites and the EOs of *C. winterianus* have demonstrated a broad range of antimicrobial activities against different pathogens [11,12,49,50]. The leaf part of *C. winterianus* EO showed effectiveness against *Pseudomonas aeruginosa*, with an MIC of 78.1 μg/mL and noticeable activity against *Staphylococcus aureus*, *Staphylococcus epidermidis*, and *Streptococcus pyogenes*, with an MIC of 156.3 μg/mL, while other panels of bacterial strains had no surprising results. The Eos of *C. winterianus* demonstrated weaker antibacterial activities than those of the positive control, gentamicin (MIC < 19.5 μg/mL). It is difficult to speculate as to which components in the root and leaves of *C. winterianus* Eos may be responsible for the antibacterial activity. In the case of *Staphylococcus aureus*, pure component citral (MIC = 78.1 μg/mL) is more active than EO. It might be due to the antagonistic effect of individual components present in EO. In the case of *Pseudomonas aeruginosa*, the leaf EO is more potent than either of the tested pure components, which might be due to the synergetic mechanism among components of EO.

The EO from the leaves of *C. winterianus* displayed potent antifungal activity against *Aspergillus niger*, *Aspergillus fumigatus*, and *Trichophyton mentagrophytes* (MIC = 78.1 μg/mL). Both the leaves and root parts of *C. winterianus* EOs showed good activity against *Candida albicans*, *Microsporum canis*, and *Trichophyton mentagrophytes*, with MIC values of 156.3μg/mL. The EOs of *C. winterianus* demonstrated weaker antifungal activities than those of the positive control, amphotericin B (MIC < 19.5 μg/mL). In the cases of *Trichophyton mentagrophytes*, *Aspergillus fumigates*, and *Aspergillus niger*, the synergetic effect is more pronounced in the leaves as compared with root EO. In the case of, *Aspergillus niger*, pure component (−) citronellal (MIC = 78.1 μg/mL) is more active than root EO. On the other hand, despite the absence of (−) citronellal, as indicated by chiral GC-MS analysis in leaves’ oil, it shows effectiveness against *Aspergillus niger*, which might be due to the synergistic effect of individual components of leaves EO. However, in the cases of *Candida albicans* and *Trichophyton rubrum*, there was no antagonistic and synergetic effect pronounced.

*C. winterianus* leaves EO showed promising antimicrobial properties and can be used in lieu of synthetic chemicals to counter microbial attacks. Additionally, the leaves of *C. winterianus* EO were more potent than the root oil. This may be due to differences in chemical compositions of EOs in roots and leaves. Alternatively, the antimicrobial properties of *C. winterianus* EO may be the presence of secondary metabolites such as citronellal, citronellol, geraniol, neral, geranial, and other components by synergistic and antagonistic mechanisms. The antifungal and antibacterial mechanisms of action of EOs are not clearly understood yet. However, it has been postulated that the hydrophobic constituents either disrupt cytoplasmic membranes via a cascade of different reactions leading to cytoplasmic leakage, cell lysis, and ultimate death, or via the inhibition of sporulation [51,52]. 

## 3. Materials and Methods

### 3.1. Plant Material and Isolation of Essential Oils

Three cultivated species of Cymbopogon, namely *C. winterianus*, *C. citratus*, and *C. martini* were collected in March, 2021 from Sunsari (26°42′19.6″ N 87°15′29.7″ E), Nepal, presented in Figure 2. Different parts of plants were separated, washed, and then hydrodistilled for 6 h using a Clevenger apparatus, as previously described [48]. The obtained EOs were dried with anhydrous sodium sulfate and stored in bottles at 4 °C until further research was conducted. The *Cymbopogon* species EOs yields are summarized in Table 1.

### 3.2. Chemical Composition Analysis by Gas Chromatography/Mass Spectrometry (GC-MS) 

Analysis of the chemical constituents in the *Cymbopogen* species (*C. winterianus*, *C. citratus*, and *C. martini*) EOs was carried out using Shimadzu GCMS-QP2010(Shimadzu Corp, Columbia, MD, USA) Ultra under the following condition: mass selective detector (MSD), operated in the EI mode (electron energy = 70 eV), with scan range = 40–400 *m/z* and scan rate of 3.0 scans/s, as previously described [53,54]. Identification of the individual components of the EOs was determined by comparison of the retention indices determined by reference to a homologous series of n-alkanes and comparison of the mass spectral fragmentation patterns (over 80% similarity match) with those reported in the literature [43] and our own in-house library [42] using the LabSolutions GC-MS solution software version 4.45 (Shimadzu Scientific Instruments, Columbia, MD, USA).

### 3.3. Enantiomeric Analysis by Chiral Gas Chromatography-Mass Spectrometry (CGC-MS)

A Shimadzu GC-MS-QP2010S with EI mode (70 eV) and B-Dex 325 chiral capillary GC column was used to perform the enantiomeric analysis of *Cymbopogen* species (*C. winterianus*, *C. citratus*, and *C. martini*) Eos, as previously described [55]. A comparison of retention times and mass spectral fragmentation patterns with authentic samples obtained from Sigma-Aldrich (Milwaukee, WI, USA) was used to identify the enantiomers. Table 5 shows the enantiomeric distribution of chiral terpenoids from *Cymbopogen* species (*C. winterianus*, *C. citratus*, and *C. martini*) EOs

### 3.4. Bacterial Strains Tested

Tryptic soy agar medium was used to culture all tested bacterial strain. A 5000 μg/mL solution of EOs was prepared in dimethyl sulfoxide (DMSO), and twofold dilution in 100 μL of cation-adjusted Mueller Hinton broth (CAMHB) (Sigma-Aldrich, St. Louis, MO, USA) was added to the top well of a 96-well microdilution plate. The prepared stock solution of EOs was then serially twofold diluted in fresh CAMHB to obtain final concentrations of 2500, 1250, 625, 312.5, 156.3, 78.1, 39.1, and 19.5 μg/mL. The freshly harvested bacteria with approximately 1.5 × 108 CFU/mL final concentration were added to each well of 96-well microdilution plates and were incubated at 37 °C for 24 h. Gentamicin (Sigma-Aldrich, St. Louis, MO, USA) and DMSO were used as positive and negative controls, respectively [56]. Seven microorganisms were used to evaluate the antibacterial activities of *C. winterianus* (leaves and root) EOs: Five Gram-positive bacteria, *Bacillus cereus* (ATCC-14579), *Staphylococcus epidermidis* (ATCC-12228), *Propionibacterium acnes* (ATCC-11827), *Staphylococcus aureus* (ATCC-29213), and *Streptococcus pyogenes* (ATCC-19615) and two Gram-negative bacteria, *Serratia marcescens* (ATCC-14756) and *Pseudomonas aeruginosa* (ATCC-27853), using the microbroth dilution technique.

### 3.5. Fungal Strains Tested

All tested fungi were cultured yeast-nitrogen base growth medium (Sigma-Aldrich, St. Louis, MO, USA). Stock solutions (5000 μg/mL) of *C. winterianus* (leaves and root) EOs were prepared in DMSO and diluted as above. The freshly harvested fungi with approximately 7.5 × 107 CFU/mL final concentrations were added to each well of 96-well microdilution plates and were incubated at 35 °C for 24 h. DMSO and amphotericin B (Sigma-Aldrich, St. Louis, MO, USA) were negative and positive antifungal controls, respectively, as previously described [56]. Seven fungal strains were used: *Aspergillus niger* (ATCC-16888), *Candida albicans* (ATCC-18804), *Microsporum canis* (ATCC-11621), *Trichophyton mentagrophytes* (ATCC-18748), *Aspergillus fumigatus* (ATCC-96918), *Microsporum gypseum* (ATCC-24102), and *Trichophyton rubrum* (ATCC-28188).

## 4. Conclusions

As far as we are aware, this is the first report on the EOs of three *Cymbopogon* species (*C. winterianus*, *C. citratus,* and *C. martini*) from Sunsari, eastern Nepal that includes not only chemical composition analysis by GC-MS but also enantiomeric composition by chiral GC-MS. The results show variations in volatiles’ compositions and enantiomeric distributions of chiral terpenoids. The yield of extraction varies depending upon the part used. The study can be used to create a benchmark for future *Cymbopogon* species’ EOs assessments, as well as authentication for adulteration or consumer safety. In addition, the antibacterial and antifungal activity of some selected pure compounds and leaves of *Cymbopogon winterianus* EO (rich in citral, citronellal, citronellol, and geraniol) suggests that it can be used in lieu of synthetic antimicrobial agents. It is unclear which of the individual components is responsible for the antimicrobial activity. However, it is likely that synergistic effects are more pronounced for the components’ activity.

## Figures and Tables

**Figure 1 molecules-28-00543-f001:**
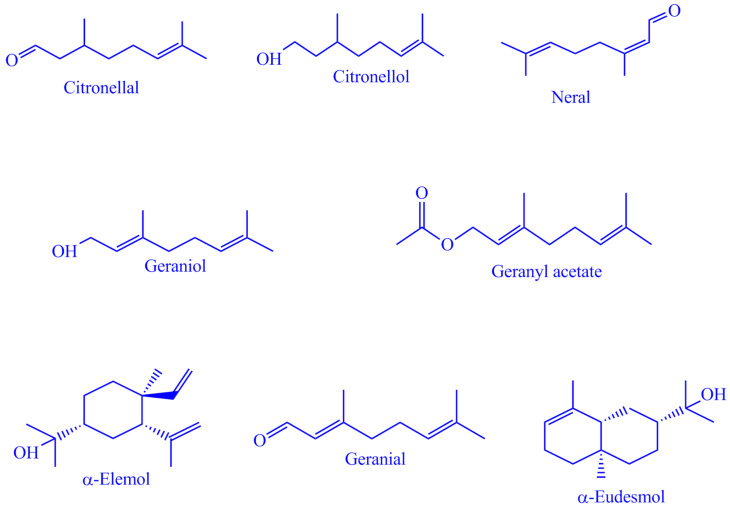
Major compounds present in different *Cymbopogon* species.

**Figure 2 molecules-28-00543-f002:**
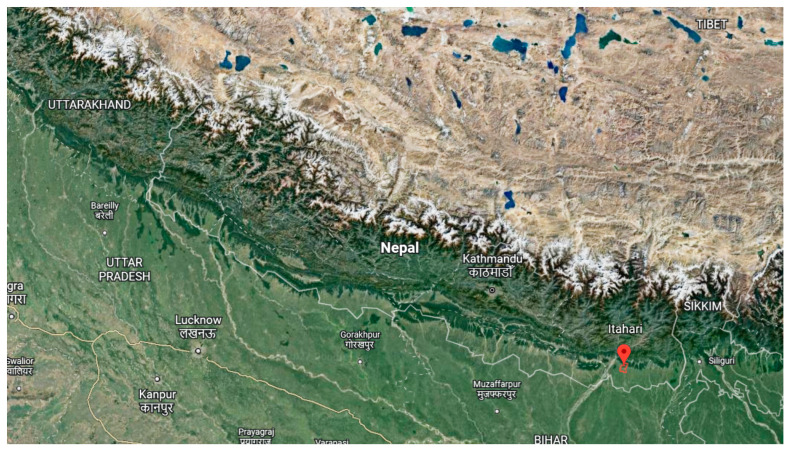
The geographical location of *Cymbopogon* species collection site from Google Earth.

**Table 1 molecules-28-00543-t001:** Essential oil yields for different *Cymbopogon* species from hydro-distillation.

*Cymbopogon* Species	Different Parts	Essential Oil Yield (*v*/*w* %)	Color of Essential Oil
*C. winterianus*	Root	0.3	Pale yellow
	Root hair and stalk (1:1)	0.3	Pale yellow
	Leaves	2.2	Light pale yellow
	Root stalk and shoot (1:1)	0.8	Pale yellow
*C. citratus*	Root	0.5	Pale yellow
	Leaves	1.4	Yellow
*C. martini*	Root	0.2	Pale yellow
	Leaves	1.3	Light pale yellow

**Table 2 molecules-28-00543-t002:** Selected constituents of *C. winterianus* essential oils from different parts.

RI_db_	RI_calc_	Compounds	Root	Root Hair and Stalk	Leaves	Root Stalk and Shoot
981	981	6-Methyl-5-hepten-2-one	t	0.1	0.2	0.8
1098	1095	Linalool	0.1	0.5	0.5	0.8
1151	1148	Citronellal	1.1	8.6	13.7	0.3
1226	1223	Citronellol	4.3	7.5	5.1	0.2
1238	1235	Neral	1.2	6.5	8.3	31.5
1250	1249	Geraniol	0.1	8.6	37.1	0.4
1268	1264	Geranial	1.9	10.7	11.0	59.0
1347	1350	Citronellyl acetate	t	1.9	1.9	-
1350	1356	Eugenol	0.7	1.2	1.0	-
1376	1379	Geranyl acetate	0.1	5.3	11.3	0.3
1387	1389	*trans*-β-Elemene	2.1	0.9	0.3	t
1417	1417	(*E*)-β-Caryophyllene	0.3	0.6	2.0	0.2
1444	1448	(*E*)-Isoeugenol	0.1	0.3	0.5	0.9
1509	1513	γ-Cadinene	0.7	0.7	0.7	0.1
1548	1548	α-Elemol	53.1	29.5	1.9	1.2
1630	1630	γ-Eudesmol	7.5	2.0	0.1	t
1659	1652	α-Eudesmol	18.9	5.3	-	-

RI_calc_ = Retention index determined with respect to homologous series of n-alkanes on a ZB-5 column. RI_db_ = Retention index from the database [42,43]. “t” indicates trace (≤0.05%) and “-” indicates not detected.

**Table 3 molecules-28-00543-t003:** Selected constituents of *C. citratus* essential oils extracted from root and leaves.

RI_db_	RI_calc_	Compounds	*C. citratus* (Lemongrass)	
			Root	Leaves
1098	1095	Linalool	0.5	0.6
1178	1177	*trans*-Isocitral	0.4	0.4
1230	1232	Epoxy geranial	0.1	0.1
1238	1235	Neral	16.6	36.1
1250	1249	Geraniol	0.4	0.3
1268	1264	Geranial	25.0	53.1
1353	1358	Neric acid	0.3	0.6
1387	1389	*trans*-β-Elemene	1.4	0.1
1417	1417	(*E*)-β-Caryophyllene	0.2	1.0
1548	1548	α-Elemol	31.5	0.3
1577	1582	Caryophyllene oxide	t	1.3
1630	1630	γ-Eudesmol	5.2	t
1659	1652	α-Eudesmol	11.3	-
1663	1666	14-Hydroxy-*epi*-(*Z*)-caryophyllene	1.0	0.1

RI_calc_ = Retention index determined with respect to homologous series of n-alkanes on a ZB-5 column. RI_db_ = Retention index from the database [42,43]. “t” indicates trace (≤0.05%) and “-” indicates not detected.

**Table 4 molecules-28-00543-t004:** Selected constituents of *C. martini* essential oils extracted from roots and leaves.

RIdb	RI_calc_	Compounds	*C. martini* (Palmarosa)	
			Root	Leaves
987	988	Myrcene	0.1	0.2
1044	1044	*trans*-β-Ocimene	0.2	0.9
1098	1095	Linalool	3.2	2.0
1238	1235	Neral	t	0.2
1250	1249	Geraniol	87.9	76.6
1268	1264	Geranial	0.2	0.6
1376	1379	Geranyl acetate	4.4	15.2
1417	1418	(*E*)-β-Caryophyllene	0.4	0.5
1713	1714	(2*E*,6*Z*)-Farnesol	3.0	1.3

RI_calc_ = Retention index determined with respect to homologous series of n-alkanes on a ZB-5 column. RI_db_ = Retention index from the database [42,43]. “t” indicates trace (≤0.05%).

**Table 5 molecules-28-00543-t005:** Enantiomeric distributions of chiral terpenoids present in *Cymbopogon* species.

Chiral Compounds	*C. winterianus* (Citronella)			*C. citratus* (Lemongrass)		*C. martini* (Palmarosa)	
	Root	Root Stalk and Shoot	Leaves	Root	Leaves	Root	Leaves
Linalool	(+) 13.0: (−) 87.0	(+) 32.4: (−) 67.6	(+) 28.7: (−)71.3	(+) 25.0: (−)75.0	(+) 33.2: (−) 66.8	(+) 95.9: (−) 4.1	(+) 86.2: (−) 13.8
Citronellal	(+) 100.0: (−) 0.0	(+) 100.0: (−) 0.0	(+) 100.0: (−)0.0	(+) 100.0: (−)0.0	-	-	-
Terpinen-4-ol	(+) 35.7: (−) 64.3	-	-	-	-	-	-
Bornyl acetate	(+) 0.0: (−) 100.0	-	-	-	-	-	-
Borneol	(+) 0.0: (−) 100.0	(+) 0.0: (−) 100	-	-	-	-	-
α-Terpineol	(+) 10.3: (−) 89.7	-	-	-	-	-	-
Citronellol	(+) 83.7: (−)16.3	(+) 54.2: (−) 45.8	(+) 83.6: (−) 16.4	(+) 60.2: (−) 39.8	-	-	-
(*E*)-β-Caryophyllene	(+) 0.0: (−) 100.0	(+) 0.0: (−) 100.0	(+) 0.0: (−) 100.0	(+) 0.0: (−) 100.0	(+) 0.0: (−) 100.0	(+) 0.0: (−) 100.0	(+) 0.0: (−) 100.0
Germacrene D	(+) 17.4: (−) 82.6	(+) 100.0: (−) 0.0	(+) 75.0: (−) 25.0	-	(+) 100.0: (−) 0.0	-	-

“-” indicates not detected.

**Table 6 molecules-28-00543-t006:** Antimicrobial activity of *C. winterianus* essential oil and some major compounds.

Name of Bacteria	*C. winterianus*		MICs (μg/mL)				
	Leaves	Root	(±)-Citronellol	Citral (Neral:Geranial in 1:1 Ratio)	(−) Citronellal	Geraniol	Gentamicin
*Bacillus cereus*	312.5	312.5	312.5	156.3	312.5	312.5	<19.5
*Propionibacterium acnes*	312.5	312.5	625	312.5	312.5	625	<19.5
*Pseudomonas aeruginosa*	78.1	2500	312.5	312.5	312.5	312.5	<19.5
*Serratia marcescens*	625	625	312.5	312.5	312.5	312.5	<19.5
*Staphylococcus aureus*	312.5	156.3	312.5	78.1	312.5	312.5	<19.5
*Staphylococcus epidermidis*	312.5	156.3	312.5	312.5	312.5	312.5	<19.5
*Streptococcus pyogenes*	625	156.3	625	156.3	312.5	312.5	<19.5
**Name of Fungus**							**Amphotericin B**
*Aspergillus niger*	78.1	2500	156.3	156.3	78.1	156.3	<19.5
*Aspergillus fumigatus*	78.1	2500	156.3	156.3	156.3	312.5	<19.5
*Candida albicans*	156.3	156.3	156.3	156.3	156.3	156.3	<19.5
*Microsporum canis*	156.3	156.3	312.5	156.3	312.5	312.5	<19.5
*Microsporum gypseum*	312.5	625	156.3	156.3	312.5	156.3	<19.5
*Trichophyton mentagrophytes*	78.1	156.3	156.3	312.5	312.5	625	<19.5
*Trichophyton rubrum*	312.5	312.5	312.5	312.5	312.5	312.5	<19.5

## Data Availability

Data included in the article/Appendix A are referenced in the article.

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
