# Peer review of "Essential Oil Composition Analysis of Cymbopogon Species from Eastern Nepal by GC-MS and Chiral GC-MS, and Antimicrobial Activity of Some Major Compounds"

_molecules, 2023, doi:10.3390/molecules28020543_

Round 1

Reviewer 1 Report

This manuscript entitled " Chemical Composition Analysis of Cymbopogon Species from Eastern Nepal by GC-MS and Chiral GC-MS, and Antimicrobial Activity of Some Major Compounds " was written in a sound English language that is almost free from linguistic and scientific errors and surveyed Cymbopogon species (C. winterianus, C. citratus, and C. martini) from Eastern Nepal chemical compounds and their antimicrobial activities against five Gram-positive bacteria, two Gram-negative bacteria and seven fungal strains.

The authors used relevant references in the research point, which added a distinctive value to the manuscript's content and the state of the art of the scientific discipline in this field. The manuscript included 51 references, including 18 modern research published during the last five years, with 5 most recent references published during this year 2022.

The introduction:

The introduction reviewed the background and economic importance of Cymbopogon species and its distribution worldwide. It also listed the damage of algae to the environment of coral reefs and their effects on growth 39 references.

Results and Discussion:

The discussion of the results was carried out correctly, in tables only, and no other method was used to present the data. The comparison with similar studies was minimal and did not match the presented results, and no statistical analyzes were made to show the differences between the plants under study. Also, the coordinates of collecting samples were not clarified or illustrated on proper geographical maps. All tables are missing the standard error and statistical significance analysis values.

In addition, the season of sampling was not determined and was more than one season compared.

The final decision is to accept this manuscript for publishing AFTER major revisions

Author Response

Reviewer I
This manuscript entitled "Chemical Composition Analysis of Cymbopogon Species from Eastern
Nepal by GC-MS and Chiral GC-MS, and Antimicrobial Activity of Some Major Compounds "
was written in a sound English language that is almost free from linguistic and scientific errors
and surveyed Cymbopogon species (C. winterianus, C. citratus, and C. martini) from Eastern
Nepal chemical compounds and their antimicrobial activities against five Gram-positive
bacteria, two Gram-negative bacteria and seven fungal strains.
Response: Thank you very much for your nice compliment. The title of our manuscript was
changed and it is “Essential Oil Composition Analysis of Cymbopogon Species from Eastern
Nepal by GC-MS and Chiral GC-MS, and Antimicrobial Activity of Some Major Compounds”.
The authors used relevant references in the research point, which added a distinctive value to the
manuscript's content and the state of the art of the scientific discipline in this field. The
manuscript included 51 references, including 18 modern research published during the last five
years, and with 5 most recent references published during this year 2022.
Response: Thank you very much for your nice compliment and highly appreciate. Some relevant
reference were added in the revised manuscript in results and discussion section to support our
existing results.
The introduction reviewed the background and economic importance of Cymbopogon species
and its distribution worldwide. It also listed the damage of algae to the environment of coral
reefs and their effects on growth 39 references.
Response: Thank you very much for your nice compliment.
The discussion of the results was carried out correctly, in tables only, and no other method was
used to present the data. The comparison with similar studies was minimal and did not match the
presented results, and no statistical analyzes were made to show the differences between the
plants under study. Also, the coordinates of collecting samples were not clarified or illustrated on
proper geographical maps. All tables are missing the standard error and statistical significance
analysis values. In addition, the season of sampling was not determined and was more than one
season compared.

Response: In this study we are trying to explore the component of different part or combined
different part of Cymbopogon species essential oil, so we discussed only in the table format. We
have added some relevant data in the results and discussion section in revised manuscript to
support existing results. There was single run of essential oil sample in GC-MS, so we have no
multiple data of single essential oil to calculate the standard deviation value for each component.
Coordinates of collecting samples sites and time has been added in the materials and method
section and the geographical location of Cymbopogon species collection site presented in the
Figure 2. To the best of our knowledge this is the first paper on Cymbopogon species and their
different part so, less data was available. There was no data available to compare in case of
combined tissue essential oil of Cymbopogon species. This research provides the data to
compared for adulteration detection, and provide a baseline for quality assessments in near
future.

Reviewer 2 Report

The manuscript "Chemical Composition Analysis of Cymbopogon Species from Eastern Nepal by GC-MS and Chiral GC-MS, and Antimicrobial Activity of Some Major Compounds" reports on the essential oils compositions of different parts of 3 Cymbopogon species and their antimicrobial activity.

The manuscript has scientific merits but many points require high attention by the authors.

Among these points:

1- The collection  time and the  color of the yields should be added since the collection time might affect the results 

2- in the chemical composition tables; The authors should add a column with the reported RIs to facilitate the validation of the results 

3- The number of replica for the GC analyses should be clearly stated

4- All the chromatograms should be given in the supplementary file

5- The reason for the chiral GC analysis is not clear and the impact on the results should be clarified 

6- Antimicrobial work without using reference antibiotics has very limited impact 

Author Response

Reviewer II
1- The collection time and the color of the yields should be added since the collection time might
affect the results.
Response: Clarification has been added to section 3.1 and color of the yield has been presented
in Table 1.
2- in the chemical composition tables; The authors should add a column with the reported RIs to
facilitate the validation of the results
Response: We have added the both literature and experimental retentions indices in revised
manuscript.  
3- The number of replica for the GC analyses should be clearly stated
Response: There was single run of essential oil sample in GC-MS, so we have no multiple data
of single essential oil to calculate the standard deviation value for each component.
4- All the chromatograms should be given in the supplementary file
Response: Due to privacy concern of testing authority, we are unable to put our data online.
5- The reason for the chiral GC analysis is not clear and the impact on the results should be
clarified
Response: As we know that essential oil is expensive due to limited natural resources. So there
is always chance of adulteration. It has been a huge challenge for researcher to authenticate
Cymbopogon species essential oil because of the use of similar natural essential oil in
adulteration. So, to the best of our knowledge this is the first research to use chiral gas
chromatography-mass spectrometry for authentication of Cymbopogon species essential oil.
Thus, by the use of chiral GC-MS we can easily detect addition of natural essential oil with
similar composition in Cymbopogon species essential oil by comparing the enantiomeric
distribution of chiral terpenoids. Previously, we published the articles “Comparison of Volatile
Constituents Present in Commercial and Lab-Distilled Frankincense (Boswellia carteri) Essential
Oils for Authentication” and in this articles we detected the different degree of adulteration with

the help of chiral GC-MS. We can easily have detected the different class of adulteration by
observing the enantiomeric ratio change in the chiral terpenoids.
6- Antimicrobial work without using reference antibiotics has very limited impact.
Response: We have clearly presented reference antibiotics in the table 6.

Round 2

Reviewer 1 Report

Thank you for your reply and edits in the manuscript

Reviewer 2 Report

The authors responded positively with most of the points 

I have no more comments